# Supervisors' ethical leadership and graduate students' attitudes toward academic misconduct

**Guangxi Zhang**[1], **Tingting Zhang**[2]*, **Sunfan Mao**[1], **Qiang Xu**[1], **Xiaoqin Ma**[1]

**1** Department of Management, Zhejiang University of Technology, Hangzhou, China, **2** School of Business, China University of Political Science and Law, Beijing, China

* zhangtingting02@cupl.edu.cn

**Data Availability Statement:** All relevant data can be accessed from OSF at http://doi.org/10.17605/OSF.IO/F7U2Q.

**Funding:** This research was funded by the Humanities and Social Science Project of the

## Abstract

Graduate students' academic misconduct has received increasing attention. Although past literature has emphasized university faculty as an important influencing factor on students' moral behaviors, the mechanisms must be further disclosed. We investigated how supervisors' ethical leadership influenced graduate students' attitudes toward academic misconduct. We explained why and how supervisor gender affects post-graduate students' social learning process by integrating social cognitive theory and role congruity theory. Study 1 used a sample of 301 graduate students in 60 academic teams in four Chinese business schools. Study 2 used experimental vignette methodology to enhance the findings' internal and external validity and provided evidence of causality. Based on the two complementary studies, we found that supervisors' ethical leadership significantly inhibited students' acceptance of academic misconduct through students' moral efficacy and the ethical climate of the academic team. The indirect effect via moral efficacy was more significant s for female supervisors. Implications for ethical leadership, academic misconduct, gender differences in leadership, and moral education were discussed.

## Introduction

Today's graduate students are the elites of tomorrow's society; thus, following academic norms is critical to their growth and success. Hoverer, according to a survey conducted by McCabe et al. [1], 43% of graduate students admitted academic cheating. Based on a sample of more than 5,000 graduate students in the United States, McCabe et al. [2] found that graduate students in business schools cheated more than peers in other schools. In a large-scale sample of 1818 students in a French business school, 70.5% reported that they cheated at least once. Simkin and McLeod [3] found that 60% of business students admitted to cheating. These findings are shocking because the immoral tendency in the college stage will continue to the work stage [4]. Business school students will potentially become organizational leaders whose unethical behaviors will damage the organization's sustainable development.

Academic misconduct is a vital research field in higher education. It is the "fabrication, falsification, or plagiarism in proposing, performing, or reporting research" [5, p. 584]. The rapid

Ministry of Education (grant number:
22YJAZH143, 22YJC630206, 22YJC630168,
22YJC630175), Zhejiang Soft Science Project
(grant number: 2022C25023), Fundamental
Scientific Research Project of Zhejiang University
of Technology for Humanities and Social Science
(grant number: GB202103005), and the Cultivation
Project of Unique Research in Humanities and
Social Sciences of Zhejiang University of
Technology (grant number: SKY-ZX-20200300).
Those grants were awarded to Guangxi Zhang.

**Competing interests:** The authors have declared
that no competing interests exist.

development of digitization makes it easier for students to obtain various materials; therefore, plagiarism has become more accessible. At the same time, the increasing pressure of publication and competition promotes academic misconduct, especially for business researchers [6]. Because academic misconduct is a sensitive and humiliating issue, some people may keep silent or rationalize the wrongdoing [7]. Thus the actual probability of academic misconduct may be higher than that reported.

Although there are multiple reasons why graduate students fail to follow academic norms [8], our understanding of the mechanisms that lead to academic misconduct still needs to be improved. Because of the difficulties in directly measuring academic misconduct, our study focused on academic misconduct attitudes. Due to the sensitivity and concealment of actual academic misconduct, we limited our analysis to attitudes towards academic misconduct, that is, the level of acceptance of such conduct. Past studies have shown that students' attitudes toward academic misconduct are strongly related to actual behaviors [3, 4, 9]. Specifically, a supervisor is essential in a graduate program to ensure that graduate students conduct meaningful and ethical research [10]. Simkin and McLeod [3] pointed out that among non-cheaters, the influence of professors was the most crucial factor in prohibiting academic cheating.

However, previous studies have not clearly explained graduate students' social learning mechanism and boundary conditions of supervisors' ethical role models. This is a significant theoretical gap because understanding how supervisors affect graduate students' attitudes toward academic misconduct provides mentorship and moral education guidelines in high education. Since the relationship between supervisors and graduate students in universities is similar to that between supervisors and subordinates in other organizations, we borrowed the concept of *ethical leadership* to describe the influence of a supervisor's ethical role model on his/her graduate students. Ethical leadership is 'the demonstration of normatively appropriate conduct through personal actions and interpersonal relationships and the promotion of such conduct to followers through two-way communication, reinforcement, and decision-making [11, p.120].

The current study reveals *how* supervisors' ethical leadership impacts graduate students' attitudes toward academic misconduct. Because unethical behaviors in business fields are relatively remarkable, we limit the analyses to graduate students in business schools. According to social learning theory [12], supervisors' ethical behaviors provide role models for students' social learning. We propose that supervisors' ethical leadership discourages graduate students' academic conduct through two paths. At the individual level, ethical leadership prevents attitudes toward academic misconduct by increasing students' moral efficacy. At the team level, ethical leadership inhibits attitudes toward academic misconduct by improving the ethical climate. Past studies indicated that leader attributes might affect observers' social learning from ethical leadership behaviors [13]. Gender is one of the most basic demographic characteristics describing leadership traits. Through integrating social cognitive theory and role congruity theory, we explore how supervisor gender plays the role of a boundary condition. Fig 1 presents our research model.

We seek to contribute to the literature in three ways. Firstly, we bridge the relatively separate literature on ethical leadership and academic misconduct. Our research represents the first empirical study investigating how supervisors' ethical leadership influences graduate students' attitudes toward academic misconduct through a multilevel research design. Secondly, according to role congruity theory [14], a supervisor's gender influences subordinates' responses to his/her leadership style, but gender roles in ethical leadership perceptions have rarely been discussed. We found that supervisor gender constitutes a boundary condition of the effect of ethical leadership, thus providing a nuanced understanding of ethical leadership and enriching the research on gender differences in leadership. Thirdly, scholars usually research ethical leadership in for-profit organizations. By extending the concept of ethical

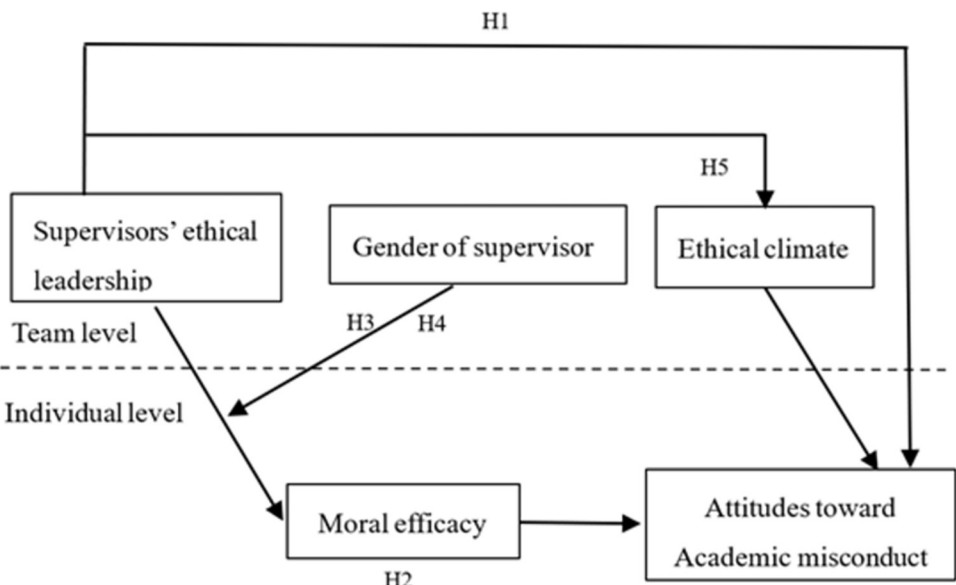

**Fig 1. A multilevel mediation model of ethical leadership to academic misconduct.**

leadership to the academic setting, we enlarge the scope of applying ethical leadership and increase its explanatory power.

## Theory and hypotheses

### Supervisors' ethical leadership and graduate students' attitudes towards academic misconduct

Ethical leaders are moral individuals and moral managers [15]. In organizations, ethical leaders establish and implement ethical standards and encourage employees' ethical behaviors through rewards and punishments. Based on social cognitive theory [12], observers visualize the behaviors of the role model and retain the visualization in their memory through observing and imitating ethical role models. When the observers' motivation is enacted under certain circumstances, the behavior preserved in the memory will become the observers' behavior.

Applying the concept of ethical leadership in the academic context, we infer that supervisors' ethical leadership impacts graduate students' attitudes toward academic misconduct. Supervisors guide graduate students in scientific research activities and provide guidance and support in students' daily studies and life. In such a master-apprentice relationship, a supervisor is embedded in the team's center and becomes a role model for students to imitate. Supervisors' ethical behaviors include setting moral examples, consistent words, and deeds, establishing academic standards and responsibilities, adhering to academic principles, and being honest and trustworthy. Graduate students may form the proper attitudes toward academic activities by observing and imitating the moral behaviors of their supervisors [16]. Supervisors' ethical leadership enhances students' understanding of academic norms and encourages students' moral identity and academic citizenship behaviors [17]. Previous studies found that professors motivated students to engage in academic integrity by trying to demonstrate ethical behaviors [18]. Therefore, we propose the following hypothesis:

*H 1*: *Supervisors' ethical leadership reduces graduate students' acceptance of academic misconduct.*

## The mediating role of the moral efficacy

According to social cognitive theory, simply "believing you can" can generate powerful effects in our life [19]. Moral efficacy is a concept extended from self-efficacy, defined as an individual's belief in his/her capabilities to organize and mobilize a series of resources to achieve moral goals when facing moral dilemmas [20].

Applying the tenets of moral efficacy in the context of higher education, we infer that supervisors' ethical leadership enhances the moral efficacy of graduate students. In academic teams, the supervisor is one of the most influential people who guide graduate students in academic activities and personal development. Firstly, supervisors' ethical leadership helps students master the strategies and rules for dealing with moral problems. Supervisors help graduate students develop moral decision-making skills through precepts and examples. Secondly, ethical leadership strengthens the psychological safety perceived by students. Admittedly, academic activities are full of challenges and uncertainties, which cause anxiety and stress concerning the outcomes. A psychologically safe environment protects students' limited cognitive and emotional resources, thus increasing students' moral efficacy. Empirical studies have supported the positive relationship between ethical leadership and subordinates' moral efficacy [21, 22].

As social learning theory states, an individual's efficacy beliefs provide a powerful motivation to engage in appropriate behaviors. Moral efficacy increases individuals' confidence in implementing different methods to solve moral dilemmas, thereby avoiding unethical behavior [20]. Schaubroeck et al. [23] uncovered that individuals with high moral efficacy tended to behave ethically when facing an ethical dilemma. Similarly, moral efficiency enables students to be more confident in solving ethical problems, increasing their awareness and motivation to avoid academic misconduct. Therefore, we propose the following hypothesis:

H2: *Moral efficacy mediates the relationship between supervisors' ethical leadership and graduate students' attitudes toward academic misconduct.*

## Integration of social cognitive theory and role congruity theory: The moderating effect of supervisor gender

Social learning is divided into four stages: Attention, representation, behavioral production, and motivation [24]. In the first stage, the attentional process is significantly affected by the characteristics of the actors. Previous studies found that leaders' traits affect subordinates' social learning of ethical leadership, such as Machiavellianism [25] and role modeling strength [26]. However, leader gender has rarely been investigated.

According to role congruity theory [14], our society has different expectations for individuals of different genders. Gender roles are the consensual beliefs about the attributes of women and men, which constitute the basis of gender stereotypes deeply rooted in our society. Therefore, a leader's gender may affect the attractiveness and effectiveness of a particular leadership style on subordinates [27]. We infer that when supervisors of different genders demonstrate ethical leadership behaviors, students' perceptions and evaluations are different, and therefore the social learning outcomes are different.

Gender stereotypes affect peoples' basic understanding of what is appropriate for male and female behaviors. Fiske et al. [28] proposed that women are generally considered to be socially oriented, with characteristics such as friendliness, selflessness, caring for others, and emotional expression. Men are competitive and achievement-oriented, with independence, assertiveness, self-confidence, and aggressiveness [29]. According to role congruity theory [14], there is consistency or inconsistency between leadership styles and genders. Because of differences in

gender role expectations [30], graduate students may expect female supervisors to show warm behaviors (including morality) and male supervisors to show dominance and capabilities.

Whether the consistency (or inconsistency) between the observed leadership style and the gender roles is attributed to leaders' dispositional factors is a decisive factor [27]. Specifically, the social roles of a male are expected to be instrumental and aggressive [31]. Thus, when male supervisors show ethical leadership behaviors, graduate students may attribute motives of ethical leadership to factors such as obtaining resources from students, gaining status, and organizational praise, as ethics, is a path to high status [32, 33]. However, perceiving that instrumental incentives drive ethical leadership will make observers question the sincerity of moral behaviors [34], thus reducing the attractiveness of ethical leadership and weakening its positive influence.

By contrast, the social roles of females emphasize warmth. Females tend to make moral judgments based on how much they care about others [35]. Thus they are perceived to be more ethical and compassionate than their male counterparts [36]. In addition, compared with males, females demonstrate a greater aversion to unethical behavior [37, 38]. Accordingly, ethical leadership behaviors of female supervisors are more likely to be perceived as within the gender role. Therefore they are more likely to be attributed to dispositional factors. As indicated in attribution principles [39], ethical leadership driven by intrinsic factors is perceived to be more sincere and attractive than that driven by extrinsic factors [40–42]. Therefore, the ethical leadership of female supervisors generates a more substantial effect on graduate students' moral efficacy than male supervisors.

*H3*: *The relationship between ethical leadership and graduate students' moral efficacy is stronger for female than male supervisors.*

Combining Hypotheses 2 and 3, we expect gender to moderate the indirect effect of ethical leadership on graduate students' attitudes toward academic misconduct via moral efficacy.

*H4*: *The indirect effect of ethical leadership on students' attitudes towards academic misconduct via moral efficacy is moderated by supervisors' gender.*

## The mediating role of team ethical climate

Team ethical climate is group members' cognitive and behavioral perceptions concerning what constitutes ethical behaviors and approaches to addressing and resolving ethical issues [43]. A team ethical climate is developed through the interaction between members and leaders and between members and members [44]. Based on social cognitive theory, ethical leadership is vital in cultivating an ethical climate in academic teams [45, 46]. Firstly, ethical leadership helps graduate students perceive the ethical climate of the universities by implementing a series of ethical standards and codes of conduct [47]. Secondly, graduate students learn and imitate the values conveyed by ethical supervisors, which develop into consistent moral norms at the team level [48].

Ethical behaviors are encouraged and rewarded in teams with a strong ethical climate. Previous studies have revealed a negative relationship between ethical climate and unethical behaviors [49]. Barnett and Voices [50] proposed that organizations can help members make correct choices when facing moral dilemmas by strengthening the organizational ethical climate. Birtch and Chiang [51] found that business school undergraduates' perception of ethical climate is a significant predictor of preventing unethical behaviors. Recently, Kuenzi et al. [52] uncovered that ethical climate negatively affects work units' unethical behaviors. Accordingly, graduate students will conform to academic norms in an academic team with a solid moral

atmosphere and less tolerance for academic misconduct. Therefore, we propose the following hypothesis:

*H5*: *Ethical climate mediates the relationship between supervisors' ethical leadership and graduate students' attitudes toward academic misconduct.*

## Study 1

### Sample and participants

We collected data from graduate students enrolled in business schools in four research universities in Eastern China. We limit the sample to business schools because academic misconduct in business schools has attracted widespread concerns, and in this way, we can control the influences of significant differences. We limited the scope of the academic team we investigated to doctoral students, academic masters, and professional masters. The primary responsibility of the supervisor was to provide academic guidance. Their qualification as a supervisor was assessed annually according to their academic achievements. A total of 303 questionnaires were distributed. To ensure that individual-level variables can be accurately aggregated to the team level, at least three members of an academic team should participate in the survey. A response rate higher than 60% in each team was ensured [53].

Participants were encouraged to provide answers honestly. All the questionnaires were filled out anonymously, and two with obvious errors were excluded. Finally, 301 questionnaires from 60 teams were used for analysis. The response rate of our survey was 99.34%. At the team level, there were 14 female and 46 male supervisors. The average number of students in the academic team was 7.60, with a minimum size of 3 and a maximum size of 15. At the individual level, there were 90 male and 211 female graduate students.

### Measures

All the continuous variables were assessed on Likert-type scales, from 1 to 7, representing 'completely disagree/unacceptable' to 'completely agree/acceptable.'

**Attitudes toward academic misconduct.** As academic misconduct is a sensitive issue that may stimulate individual self-defense, we used anonymous answers and projective techniques. We adopted the six-item scale developed by Zhang and Yu [54]. This scale was originally developed to measure the academic misconduct of Chinese graduate students. Items include: "adjust or modify data," "citing unread literature in references," "publishing translated foreign literature," "submitting papers to more than one journal," "quoting others' opinions without adding notes," and "plagiarizing others' research work" (Cronbach's alpha ($\alpha$) = 0.85). We asked graduate students to judge the extent to which they would consider these behaviors acceptable if other students exhibited them.

**Ethical leadership.** We used the 10-item scale Brown et al. [11] developed to measure ethical leadership. The wordings were modified to make the expressions appropriate for an academic context. Sample items included 'my supervisor conducts their personal life ethically ($\alpha$ = 0.93).

The mean $r_{wg}$ value for ethical leadership was 0.86. The ICC1 value was 0.20, thus suggesting strong agreement within teams regarding ethical leadership and the appropriateness of aggregating individual responses to the group level [55]. The ICC2 value was 0.53.

**Ethical efficacy.** The 5-item scale developed by Hannah and Avolio [56] was used to measure moral efficacy. Sample items included 'I am confident that I can determine what needs to be done when I face ethical dilemmas' ($\alpha$ = 0.86).

**Ethical climate.**   We used the scale Schwepker Jr [57] developed to measure the ethical climate. We modified the expressions to represent team-level climate. The original scale has seven items. Two inappropriate items in our research context ('our team has policies regarding ethical behavior' and 'our team enforces policies regarding ethical behavior') were deleted. Sample items included 'unethical behavior is not tolerated in our team' ($\alpha = 0.87$).

The mean within-group agreement ($r_{wg}$) value for the ethical climate was 0.92. The intraclass correlation coefficient (ICC)1 value was 0.58, thus suggesting strong agreement within work groups regarding ethical climate and the appropriateness of aggregation of individual responses to the group level (55). The ICC2 value was 0.89.

**Control variables.**   We controlled for the gender, age, and academic years of graduate students at the individual level. We controlled for team size at the group level. Gender was coded as one for males and zero for females. We coded 1 for students under 22, 2 for those 22–25, 3 for those 26–29, and 4 for those over 30. Students' academic years were the sum of years and months of participation in the educational program. Because the measured dependent variable was sensitive, social desirability bias might occur. Thus, we controlled for social desirability [58]. This scale has five items. A sample item included "would you ever lie to people?" ($\alpha = 0.73$).

## Results

We used SPSS 25.0, HLM 7.0, and MPLUS 7.0 to analyze the data. The construct validity was tested using confirmatory factor analysis (CFA), including four latent variables: supervisors' ethical leadership, attitudes toward academic misconduct, ethical climate, and moral efficacy. The hypothesized measurement model showed a good overall model fit ($\chi^2_{(293)} = 799.89$, RMSEA = 0.07, CFI = 0.95, NNFI = 0.91). All the standardized loadings were significant, ranging from 0.67 to 0.94 ($p < 0.001$). The composite reliability of those constructs exceeded the 0.70 benchmark, and the average variances extracted (AVE) were greater than 0.50. We also compared the hypothesized four-factor model with other likely models, and none of the alternative models had a better fit than the baseline model. Therefore, these findings confirmed the convergent and discriminative validities of our measures.

**Common method variance (CMV).**   As the data were derived from self-reports, common method bias might exist [59]. We ensured that the common method variance would not affect the accuracy of the results through ex-ante prevention and post-hoc analyses. In the ex-ante prevention, we used anonymous answers, an indirect questioning technique, and included social desirability as a control variable. We applied Harman's single-factor test to rule out the common method variance in the post-hoc analyses. An explanatory factor analysis (EFA) was performed on all 26 items of the four key continuous variables. The factor analysis generated four distinct factors (eigenvalue >1.0), accounting for 66.36% of the total variance, with no single factor accounting for most of the variance. The variation explained by the first principal component was 23.81%, which was less than the critical standard of 40% [59]. The method of controlling for the effect of an unmeasured latent methods factor was used to perform the test again. The result showed that, after adding an unmeasured latent methods factor, the fitting index of the five-factor model was not significantly improved than that of the four-factor model. Thus, common method variance was not a significant concern in this study.

Table 1 presents the means, standard deviations, and correlation coefficients of variables at the individual and group levels. A significant positive correlation existed between ethical leadership and ethical climate ($r = 0.61$, $p < 0.001$), and a significant negative correlation existed between moral efficacy and attitudes toward academic misconduct ($r = -0.48$, $p < 0.001$).

**Tests of hypotheses.**   In the multilevel analyses, we group-mean centered Level 1 variables.

**Table 1. Means, standard deviations, and correlations of the variables.**

| Variables | M | SD | 1 | 2 | 3 | 4 | 5 | 6 |
|---|---|---|---|---|---|---|---|---|
| *Individual level predictors* | | | | | | | | |
| 1. Moral efficacy | 4.88 | 0.76 | | | | | | |
| 2. Academic misconduct | 2.05 | 0.94 | −0.48*** | | | | | |
| 3. Academic years | 1.71 | 0.88 | 0.06 | 0.00 | | | | |
| 4. Gender of student | 0.30 | 0.46 | −0.25*** | 0.22*** | −0.04 | | | |
| 5. Age | 2.24 | 0.50 | 0.03 | −0.02 | 0.15** | 0.09 | | |
| 6. Social Desirability | 4.19 | 1.01 | 0.15** | −0.17** | 0.06 | −0.07 | 0.04 | |
| *Team level predictors* | | | | | | | | |
| 1. Ethical leadership | 5.61 | 0.57 | | | | | | |
| 2. Ethical Climate | 5.08 | 0.50 | 0.61*** | | | | | |
| 3. Gender of supervisor | 0.77 | 0.43 | −0.04 | −0.15 | | | | |
| 4. Team size | 7.60 | 3.45 | −0.26* | −0.46*** | 0.28* | | | |

**Note:** (1) ***Coefficient significant at 0.1%

** significant at 1%

* significant at 5%.

*Test of hypothesis 1*. Hypothesis 1 proposes that supervisors' ethical leadership restrains graduate students' acceptance of academic misconduct. As Model 4 in Table 2 showed that the effect of ethical leadership on attitudes toward academic misconduct was significant ($r = -0.36$, $p < 0.01$). Thus, Hypothesis 1 was supported.

*The mediating role of moral efficacy*. Hypothesis 2 proposes the mediating role of moral efficacy between supervisors' ethical leadership and students' attitudes toward academic misconduct. As shown in Table 2, ethical leadership and moral efficacy were positively related ($\gamma = 0.26$, $p < 0.05$, in Model 2), and the effect of moral efficacy on attitudes towards academic misconduct was significant ($\beta = -0.55$, $p < 0.001$, in Model 5). When supervisors' ethical leadership and moral efficacy were introduced into the regression of attitudes towards academic misconduct in Model 7, the effect of supervisors' ethical leadership on attitudes towards academic misconduct maintained significant (changing from $\gamma = -0.36$, $p < 0.01$ in Model 4 to $\gamma = -0.23$, $p < 0.05$, in Model 7). Thus, moral efficacy partially mediated the relationship between supervisors' ethical leadership and attitudes toward academic misconduct. The bootstrapping approach (bootstrapping = 5000) showed that the mediating role of moral efficacy was significant ($b = -0.41$, 95% CI not containing zero). Thus, Hypothesis 2 was supported.

*The moderating effect of supervisor gender*. Hypothesis 3 proposes that the positive relationship between supervisors' ethical leadership and moral efficacy was stronger for female supervisors. As shown in Table 2, the interaction coefficient between ethical leadership and the gender of the supervisor was significant ($\gamma = -0.47$, $p < 0.05$, in Model 3). Hence gender of the supervisor significantly moderated the relationship between ethical leadership and graduate students' moral efficacy. Thus, Hypothesis 3 was supported. The indirect effect of supervisors' ethical leadership on attitudes towards academic misconduct through moral efficacy was significant when the supervisor was female ($b = -0.77$, $p < 0.01$) but not significant when the supervisor was male ($b = -0.08$, $p > 0.05$). Thus, Hypotheses 4 received support.

*The mediating role of team ethical climate*. Hypothesis 5 predicts the mediating role of team ethical climate between supervisors' ethical leadership and attitudes towards academic misconduct. As shown in Table 2, supervisors' ethical leadership was a significant predictor of ethical climate ($b = 0.42$, $p < 0.001$, in Model 1), and ethical climate significantly predicted attitudes towards academic misconduct ($\gamma = -0.63$, $p < 0.001$ in Model 6). Model 8 demonstrated that

**Table 2. Regression results using ethical climate, moral efficacy, and academic misconduct as dependent variables.**

| Variables | Ethical climate | Moral efficacy | | Academic misconduct | | | | |
|---|---|---|---|---|---|---|---|---|
| | Model 1 | Model 2 | Model 3 | Model 4 | Model 5 | Model 6 | Model 7 | Model 8 |
| **Intercept** | 4.1051*** (0.8625) | 2.8617*** (0.6555) | 0.9322*** (1.1672) | 4.8398*** (0.8039) | 4.9260*** (0.4427) | 6.4129*** (0.7975) | 6.3414*** (0.7268) | 6.7177*** (0.8575) |
| *Level 1 predictors* | | | | | | | | |
| Academic years | −0.0563 (0.1197) | 0.0485 (0.0472) | 0.0322 (0.0476) | −0.0012 (0.0592) | 0.0316 (0.0549) | −0.0031 (0.0582) | 0.0267 (0.0546) | −0.0041 (0.0582) |
| Gender | −0.4266* (0.1961) | −0.3659** (0.0922) | −0.3487*** (0.0925) | 0.3248** (0.1153) | 0.1783 (0.1090) | 0.2664* (0.1148) | 0.1677 (0.1084) | 0.2653* (0.1148) |
| Age | −0.1572 (0.1992) | 0.0411 (0.0865) | 0.0426 (0.0861) | −0.0057 (0.1079) | −0.0035 (0.0993) | −0.0887 (0.1038) | 0.0048 (0.0983) | −0.0758 (0.1046) |
| Social Desirability | −0.1040 (0.1004) | 0.1051* (0.0410) | 0.1042* (0.0409) | −0.1736** (0.0513) | −0.1084* (0.0479) | −0.1732** (0.0506) | −0.1163* (0.0479) | −0.1754** (0.0506) |
| Moral Efficacy | | | | | −0.5238*** (0.0664) | | −0.5022*** (0.0668) | |
| *Level 2 predictors* | | | | | | | | |
| Team size | −0.0442** (0.0145) | −0.0059 (0.0145) | −0.0171 (0.0151) | 0.0025 (0.0176) | 0.0051 (0.0152) | −0.0224 (0.0160) | −0.0018 (0.0147) | −0.0217 (0.0161) |
| Ethical leadership | 0.4139*** (0.0871) | 0.2809** (0.0100) | 0.6155** (0.1780) | −0.3861** (0.1222) | | | −0.2565* (0.1068) | −0.1252 (0.1276) |
| Ethical climate | | | | | | −0.6617*** (0.1235) | | −0.5874*** (0.1449) |
| Gender of supervisor | | | 2.6692* (1.3288) | | | | | |
| Supervisor's Ethical Leadership× Gender of Supervisor | | | −0.4392* (0.1960) | | | | | |
| **Pseudo $R^2$** | 0.5478 | 0.4198 | 0.4270 | 0.6820 | 0.6063 | 0.7274 | 0.6209 | 0.7299 |

**Note:** (1) Standardized regression coefficients are reported with two-tailed tests

(2) Standardized coefficients are reported; Standard errors are in brackets

(3) \*\*\*Coefficient significant at 0.1%

\*\* significant at 1%

\* significant at 5%.

when the ethical climate was entered into the equation, the significant main effect of supervisors' ethical leadership on attitudes towards academic misconduct became insignificant ($\gamma = -0.11$, *n.s.*), indicating full mediation. The bootstrapping approach (bootstrapping = 5000) showed that the mediating role of ethical climate was significant (indirect effect = −0.31, 95% CI not containing zero). Thus, Hypothesis 5 was supported.

## Study 2

Since gender was unbalanced in Study 1, we conducted a vignette study in Study 2. The vignetter study combines the advantages of experiment and survey. It is an appropriate method when the researcher's purpose is to discover causality and the research topics are sensitive [60, 61].

### Method

**Design and participants.** A total of 158 graduate students in business schools in three Chinese research universities were recruited with wenjuan.com (an online data collection

service). The questionnaires were filled out anonymously to increase data objectivity. Participants were randomly assigned to four scenarios: 2 (high vs. low ethical leadership) × 2 (male supervisor vs. female supervisor).

**Procedures and materials.** The participants were told that the questionnaires were anonymous and used for academic purposes. There were no right or wrong judgments in their answers, and thus they were encouraged to express their feelings honestly. In the first step, the participants were presented with personal information about a professor, including name, gender, position, telephone number, and e-mail. To improve the participants' immersion in the scenarios, we used texts and pictures to display the supervisor's information and created a virtual personal webpage for the supervisor. Then, participants read additional details on the scenarios which manipulated ethical leadership. Finally, they were asked to rate their attitudes toward academic misconduct if such a supervisor supervised them.

*Manipulation of ethical leadership*. Referring to approaches of manipulating ethical leadership in previous studies [62, 63], we described supervisors' ethical leadership based on the conceptualization of ethical leadership [11]. The expression was changed slightly according to the context in universities. Under the high ethical leadership condition, the participants read the following passage:

> *'He/she is virtuous in his/her work and life. He/she cares about the interests of his/her students. When facing ethical dilemmas, He/she always adheres to the right values. When doing research, he/she is not only concerned about the publication of the papers but also about whether the research process is rigorous. He/she often communicates and discusses with students and guides them to form correct values. If a student demonstrates behaviors that do not conform to academic norms in research activities, he/she will remind the student. Most students feel that he/she is a trustworthy supervisor.'*

Under the low ethical leadership condition, the participants read the following passage:

> *'His/her morality in work and life is average. He/she seldom considers the interests of students. When facing an ethical dilemma, he/she sometimes cannot adhere to the correct values. When doing research, he/she is only concerned about publishing the papers but does not care much about whether the research process is rigorous. He/she seldom communicates and discusses with students and ignores guiding students to form correct values. If a student demonstrates behaviors that do not conform to academic norms in research activities, he/she will not remind the student. Most students do not feel that he/she is a trustworthy supervisor.'*

## Measures

All the items were assessed on Likert-type scales (1 = strongly disagree/highly impossible, 7 = strongly agree/highly possible). The male was coded as 1, and the female was coded as 2.

*Manipulation check of ethical leadership*. Items were modified according to the 10-item ethical leadership scale developed by Brown et al. (11). A sample item is 'this supervisor conducts his/her personal life in an ethical manner" ($\alpha$ = 0.98).

*Attitudes towards academic misconduct*. The items measuring attitudes towards academic misconduct were the same as those used in Study 1. We asked the subjects: "If such a supervisor instructed you, to what extent would you feel the following behaviors acceptable? "A sample item is "plagiarizing others' research work" ($\alpha$ = 0.95).

## Results

**Manipulation check.** Results showed that the participants under the high ethical leadership condition gave a higher rating of ethical leadership (M = 6.08, SD = 0.82) than those in the low ethical leadership condition (M = 2.53, SD = 1.51), $t$ (156) = 18.57, $p < 0.001$.

**Tests of hypotheses.** Hypothesis 1 predicts that supervisors' ethical leadership negatively affects students' acceptance of academic misconduct. We tested the normality of the dependent variable in four groups. Due to the small sample size, the data did not meet normal distribution. We used the Mann-Whitney U test to test the differences between two independent samples. Participants under the high ethical leadership condition displayed less acceptance of academic misconduct (M = 2.35, SD = 1.48) than those under the low ethical leadership condition (M = 4.79, SD = 1.40), $Z = 8.12$, $p < 0.001$. The main effect of ethical leadership was significant: $F$ (1, 156) = 24.56, $p < 0.001$, and $\eta^2 = 0.42$. Therefore, Hypotheses 1 was supported.

Hypotheses 2 to 4 were tested using Hayes' PROCESS macro in SPSS 25.0. Hypothesis 2 predicts that moral efficacy mediates the relationship between supervisors' ethical leadership and attitudes toward academic misconduct. In support of Hypothesis 2, the indirect effect was significant: $ab = -.43$, 95%, CI = [− 0.78, − 0.08].

Hypothesis 3 predicts that supervisor gender moderates the relationship between supervisors' ethical leadership and students' moral efficacy. The interaction between ethical leadership and gender was significant: $F$ (1, 154) = 4.82, $p < 0.05$, and $\eta^2 = 0.03$. Thus, Hypothesis 3 was supported.

In support of Hypothesis 4, for female supervisors, the indirect effect of ethical leadership on students' attitudes towards academic misconduct through moral efficacy was negative and significant: $ab = - 0.56$, CI = [−.99, − 0.11]. For male supervisors, the negative indirect effect was less significant: $ab = - 0.34$, CI = [− 0.68, − 0.07]. The index of the moderated mediator was negative and significant: index = − 0.21, CI = [− 0.52, − 0.01]. Thus, Hypothesis 4 was supported.

## Discussion

The intensifying pressures are likely to increase the probability of graduate students' academic misconduct. We found that supervisors' ethical leadership significantly prohibited academic misconduct of graduate students by enhancing students' moral efficacy and the ethical climate of academic teams. By integrating social cognitive theory and role congruity theory, we disclosed that supervisor gender was a boundary condition that affected graduate students' social learning.

### Theoretical contribution

This research makes several theoretical contributions. Firstly, we enrich the antecedents and mechanisms of attitudes toward academic misconduct. A recent literature review on academic misconduct found that only 4% of studies used multiple methods [64]. Our research provides reliable knowledge for developing this field by combining cross-level questionnaire surveys and experimental research. Although past studies recognized the role of tutors in shaping students' morality [65], the processes lacked empirical testing. Our study suggests that role congruity theory can supplement and enrich the traditional explanation based on social learning theory. We also found that supervisors' ethical leadership can prohibit students' academic misconduct by enhancing students' moral efficacy and improving the ethical climate, providing a refined understanding of the effects of ethical leadership through two paths.

Secondly, we provide new insights into female leadership advantages. Admittedly, it is more difficult for women to achieve leadership positions and career success than men because

almost all the characteristics of successful leadership are described using the male framework [66]. However, women do not have to limit themselves to the framework set by men. Being female could enhance ethical leadership's positive effects on students' social learning outcomes. This finding enriches the research on female advantage in leadership. Although the mainstream view is that adopting a gender-congruent leadership style generates stronger effects [66], some studies revealed opposite findings [27]. We resolve the debate by stating that researchers need to analyze observers' attribution and the observed congruence/incongruence of gender roles.

Finally, we extend the application scope of ethical leadership. Although scholars have done many studies on the consequences of ethical leadership, most outcome variables are related to employee behaviors in the context of enterprises, such as job satisfaction and turnover intention. Therefore, we contribute new knowledge to its effects in the academic context. In addition, recent studies suggested that more than mere exposure to ethical leadership is needed to guarantee successful social learning [13]. Our findings contribute new knowledge to ethical leadership's boundary conditions in social learning.

## Practical implications

This study has practical implications for graduate students' moral education and management system. Many Chinese graduate students suffer from low academic standards and poor mentoring [67]. The continuous expansion of the enrollment scale of graduate students in Chinese universities has brought increasing pressure on publication and employment. According to a survey conducted at 10 Chinese research universities, the lack of appropriate supervision and punishment contributed to academic misconduct [68]. Supervisors' influences are especially significant in Confucian cultures, as students respect their teachers greatly. To strengthen students' morality effectively, universities should focus more on cultivating and evaluating supervisors' ethics. However, the global pursuit of university ranking pushes higher education to allocate more resources to scientific research than training students. The recruitment of academic staff is mainly based on their professional ability (i.e., doing research). To consolidate students' academic integrity in high education, universities should adjust the personal selection and assessment criteria.

Mentoring graduate students in developing countries face more ethical challenges than in developed countries, which is especially salient in business schools. With the expansion of the enrollment scale of business schools, supervisors are facing increasing pressure to guide students, and students are facing increasing challenges in following academic norms. Business schools worldwide have paid more attention to the content of business ethics in their curriculum. Although most Chinese universities have formulated relevant punishment systems for academic misconduct, it is not strictly enforced in many cases. Therefore, developing a detailed and implementable punishment system is another solution to ethical issues.

A key finding in our study is that students respond differently to ethical leadership of different genders. Since most leadership positions in our society are still occupied by males, exploring leadership styles suitable for females is conducive to improving women's career opportunities. Our research supports females' advantages in demonstrating ethical leadership. Therefore, female supervisors can take advantage of gender roles to make students feel that their ethical leadership behaviors are sincere. Male supervisors should avoid students attributing ethical leadership behaviors to instrumental motives. Therefore, male supervisors can change graduate students' perception of ethical leadership motives by strengthening dyadic communication and trust. Observers' attribution of ethical leadership motives of different genders is an intriguing issue, and we encourage future research to explore this vital direction.

### Limitation and future direction

Our study has several limitations that offer valuable opportunities for future research on this critical topic. Firstly, the perception and attribution of inconsistent gender roles are the core tenets of the arguments for the moderating effect of gender. However, we did not directly measure graduate students' perceptions of ethical leadership motives. Because the role congruity theory alone cannot wholly explain the inconsistent findings on gender roles, analyzing observers' attribution of leadership styles in conjunction with gender role congruity can provide an in-depth understanding of the mechanisms. Therefore, we hope future research can better conceptualize and measure observers' attribution process of ethical leadership behaviors of leaders of different genders.

Secondly, there is still room for further improvement in the study design. The measures of attitudes are susceptible to social desirability bias [69]. Although our research adopted the indirect questioning method and added a control variable of social desirability, those measures could only partially prevent respondents from providing socially desirable responses. Future researchers can use big data and management information systems to directly measure academic misconduct (e.g., cheating, plagiarism). In that case, the measurement of academic misconduct will be more direct and accurate.

Furthermore, one of the main areas for improvement of the vignette study is the artificial nature of the research situation. Although we adopted a combination of texts and pictures to immerse the subjects in the scenarios as much as possible, those approaches might need to be revised to generate strong incentives in experiments. To engage participants' senses more deeply, future researchers can use more advanced technologies, such as videos, games, and virtual reality technology, to let participants see and feel the appearance and behaviors of the supervisors.

Thirdly, scholars should further examine the generalizability of our results in other cultural contexts. In Study 1, we limited the sample to graduate students in business schools. Thus future research can investigate other majors to increase the ecological validity of the findings. In addition, this study was conducted in China, which is characterized by high collectivism, high power distance, and Confucian ethics [67]. For example, gender role expectations are strong in Chinese culture [70]. In China, females are expected to be feminine and compliant, while males are expected to be aggressive and masculine. Therefore, the expectation of female roles in Chinese society is very close to the warm (e.g., ethical) dimension in social recognition, but it is not necessarily so in other cultures. China has a highly collectivist culture and a tradition of respecting teachers. Therefore, compared with an individualist culture, supervisors' ethical leadership may have a more significant impact on students' ethical behaviors.

### Conclusion

Compared with developed countries, conducting high-quality research in developing countries faces more significant ethical challenges [10]. Academic misconduct wastes many resources and damages the long-term development of graduate students. To the best of our knowledge, this is the first study that examined how supervisors' ethical leadership inhibited graduate students' academic misconduct and analyzed the boundary condition. We hope that our research can inspire scholars to think more deeply about how to curb academic misconduct. Today's business school students are future business leaders, and we hope our research can help university administrators design a better education management system in business schools.

## Author Contributions

**Conceptualization:** Guangxi Zhang.

**Formal analysis:** Guangxi Zhang.

**Funding acquisition:** Guangxi Zhang.

**Methodology:** Guangxi Zhang, Sunfan Mao.

**Resources:** Qiang Xu.

**Software:** Sunfan Mao.

**Supervision:** Guangxi Zhang, Tingting Zhang, Qiang Xu.

**Writing – original draft:** Guangxi Zhang, Sunfan Mao.

**Writing – review & editing:** Guangxi Zhang, Tingting Zhang, Xiaoqin Ma.

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
