## [Decision Letter · Decision Letter 0]

5 Sep 2022

PONE-D-22-10477Supervisors' ethical leadership and academic misconduct of graduate studentsPLOS ONE

Dear Dr. Zhang,

Thank you for submitting your manuscript to PLOS ONE. After careful consideration, we feel that it has merit but does not fully meet PLOS ONE’s publication criteria as it currently stands. Therefore, we invite you to submit a revised version of the manuscript that addresses the points raised during the review process. Both reviewers and myself see merit in your endeavour, however, they highlight important issues that have to be solved  before the paper could be published. Clarifying wether attitudes or behavior are being analized is fundamental, clearly describing the variables also, and the experiment should be incentivized. If you are not able to carry out the additional incentivized sessions proposed, you should acknowledge this weakness of the study in the conclusions and abstain from calling experiment the second study, and call it survey instead. If you decide to resubmit the paper please take into account that I will send it again for re-evaluation to the same reviewers.

We look forward to receiving your revised manuscript.

Kind regards,

Iván Barreda-Tarrazona, PhD

Academic Editor

PLOS ONE

Journal Requirements:

NO authors have competing interests

4. Please amend the manuscript submission data (via Edit Submission) to include author Qiang Xu.

Reviewers' comments:

Reviewer's Responses to Questions

**Comments to the Author**

1. Is the manuscript technically sound, and do the data support the conclusions?

Reviewer #1: No

Reviewer #2: Partly

2. Has the statistical analysis been performed appropriately and rigorously? 

Reviewer #1: No

Reviewer #2: No

3. Have the authors made all data underlying the findings in their manuscript fully available?

Reviewer #1: Yes

Reviewer #2: Yes

4. Is the manuscript presented in an intelligible fashion and written in standard English?

Reviewer #1: No

Reviewer #2: Yes

5. Review Comments to the Author

Reviewer #1: (1) This paper tackles the supervisors’ (academic advisors’) ethical leadership and academic misconduct of graduate students. It is done nicely in terms of topic, theoretical foundation, subject selection, statistic method. However, there is a major flaw must be clarified: what exactly have the authors measured as the variable of “academic misconduct”? In my observation, this study only measures graduate students’ “attitudes” toward academic misconduct by asking whether they disagree/accept some misconduct types (e.g., adjust or modify data, citing unread literature in references, submitting papers to more than one journal, etc.). In other words, the authors did not actually measure students’ actual wrong doings, frequency of wrong doings, or even their behavioral intention. Therefore, all hypotheses and results, such as “H1 Supervisors’ ethical leadership inhibits academic misconduct of graduate students” (p.7), “We also found that supervisors’ ethical leadership can prohibit students’ academic misconduct through enhancing students’ moral efficacy….”(p.23) are very misleading. When the authors said “supervisors’ ethical leadership inhibits graduate students’ academic misconduct”, it should be “supervisors’ ethical leadership reduces graduate students’ agreement level (or acceptance level) of these misconducts,” not their actual misconduct committed. Although the authors considered to use “projective techniques” to alleviate the concerns of respondents, I don’t think the way items asked can justify or interpret attitudes to actual behaviors. In sum, the authors must clarify the measurement of variables and rewrite the whole article.

(2) This paper has two sets of hypotheses. First set (study 1) has five hypotheses, and the second set (study 2) has four. I would suggest to renumber (such as H1-1, 1-2, 1-3, 2-1, 2-2…) these hypotheses for better clarity. If possible, the authors may provide graph(s) to visually show the set of five hypotheses and another set of four hypotheses.

(3) In the scale measuring “Ethical climate,” please explain more on why two items were deleted from the original 7-item scale. In my view, items “our team has policies regarding ethical behavior” and “our team enforces policies regarding ethical behavior” can be modified to fit the situation of the present study, such as “our research team has integrity policies regarding academic ethical behavior.”

(4) This paper also needs a revision on its English writing. Some sentences are not complete, such as “Even the same ethics behaviors are not imitated equally by observers [10]. In theory, the social learning…” (p. 4). Some are typos or wrong use of words, such as “In high education, moral…” (p. 8), to name a few.

Reviewer #2: The paper reports a study trying to explain why and how supervisor gender and ethical leadership affects post-graduate students' social learning process. The authors adopted a multi-method approach to test the predictions in two studies. One study was a field experiment, and the second study was an experiment to enhance the robustness of the findings in the first study. The object of the analysis is presented in a convincing way and highlight its relevance. Although the paper is well written and the topic is certainly of interest, I am not sure it is addressed in the best way possible.

In the first study (Study 1), they collected data from graduate students in four Chinese business schools. A total of 301 subjects correctly answered four questionaries, with a total of 26 items to measure Academic misconduct, Supervisor’s ethical leadership, Ethical efficacy, and Ethical climate. Concerning the dataset, the authors should consider and comment a possible percentage of fake answers of the questionaries. In addition, it might be interesting to know if the supervisors are also the directors of their academic team. Moreover, the authors should describe the code for the age variable as they described the other control variables in the paper.

Concerning the methodology used in this study, the authors used parametric tests such as Pearson’s test, a method of statistical analysis that assume a normally distribution. In my opinion, it would be better to use nonparametric tests to avoid this assumption. In table 2, the authors should include the standard errors and a third or fourth decimal in all coefficients instead of writing 0.00 coefficient in model 4 and 6. Furthermore, in the title of the table 2, the regression of the ethical climate model does not appear.

As they said in the paper, given that gender was unbalanced in Study 1, they conducted a lab experiment (Study 2) to enrich their results and provide evidence of causality. However, in my opinion (if I understood correctly), the experiment was not designed as well as possible or a thorough experimental design section is needed in the paper. As a control experiment, the authors should be considered the same control variables of the subjects as in the first study. Moreover, the authors should control the number of subjects in the four groups considering high (low) ethical leadership and male (female) supervisor. In addition, in order to control the fake answers of the questionaries, an experimental design with real reward could be considered. In fact, it would be interesting to include some game that implements a punishment system as a solution to the ethical issues.

Concerning the methodology used in the study 2, again the authors used parametric tests. I would use a Mann-Whitney test instead of t test.

Minor comments:

- In page 11, in Hypothesis 3, instead of writing “Supervisor gender moderates…” I could write “Woman (or female) supervisor…”

- In my opinion, the authors should write the name of the indices that they used in the analysis of the variables. For example, The internal consistency (�), the Intraclass correlation coefficient (ICC), the measure of within-group agreement, …

- The authors should check number of cites in the text. In page 13, “…Zhang and Yu [55]…” is [56]. In page 14, “(Bliese, 2000)” is reference [57]. “The scale developed by Schwepker Jr [56]…” is [58]. Page 27, “…Calhoun, 1995…” is reference [67]

- The authors should check reference Zhang and Yu 2017.

- In page 16, “An EFA was performed…” include Explanatory Factor Analysis (EFA)

- References 65-66 are not cited in the text

My overall impression of the manuscript is that it needs a thorough revision following the comments made in the preceding lines. In particular, the lab experimental design. I rather suggest to send it a specific journal.

6. PLOS authors have the option to publish the peer review history of their article (what does this mean?). If published, this will include your full peer review and any attached files.

Reviewer #1: No

Reviewer #2: No

---

## [Author Response · Author response to Decision Letter 0]

13 Nov 2022

Supervisors' ethical leadership and academic misconduct of graduate students

PONE-D-22-10477

Executive summary of the revision

1. We appreciate those constructive comments. We have tried to improve this revised version's theoretical reasoning and research method. 

2. We have revised the expression of "academic misconduct" in the full text and expressed it more clearly as "attitudes toward academic misconduct."

3. We now call the second study as a vignette study. We discussed the advantages and disadvantages of this approach in the Discussion section.

4. We have added social desirability as a control variable to increase the validity and reliability of the results in Study 1. We have updated Tables 1 and Tables 2.

5. We have tested the data distribution in Study 1 and Study 2 to prove that our statistical methods are appropriate.

6. We follow one of the reviewers' suggestions and use the Mann-Whitney U Test for the statistical analysis in Study 2.

7. We have supplemented the information about the supervisor in the main text. We have explained why we deleted the two items in the scale of ethical climate in the response letter.

8. We have polished and proofread the writing of the full text.

9. We have corrected some minor issues.

In order to distinguish from the Vancouver Style in the text and reviews’ comments, we use AMJ style in the response letter. Protocols.io is mainly for biochemistry, molecular biology, and biomedicine. The process of study 2 was explained, and the data was uploaded online; thus, we do not deposit materials in protocols.io.

We hope that you find that this paper has been significantly improved as a result of incorporating most, if not all, of your comments. Again, we thank you for your continued guidance and support throughout the review process.

The Authors Team

Responses to Comments of Referee 1

Thank you for your detailed and constructive comments. Our responses to your comments (underlined in blue) are as follows.

R1-0 This paper tackles the supervisors’ (academic advisors’) ethical leadership and academic misconduct of graduate students. It is done nicely in terms of topic, theoretical foundation, subject selection, statistic method. 

However, there is a major flaw that must be clarified: what exactly have the authors measured as the variable of "academic misconduct"? In my observation, this study only measures graduate students' "attitudes" toward academic misconduct by asking whether they disagree/accept some misconduct types (e.g., adjust or modify data, citing unread literature in references, submitting papers to more than one journal, etc.). In other words, the authors did not actually measure students' actual wrong doings, frequency of wrong doings, or even their behavioral intention. Therefore, all hypotheses and results, such as "H1 Supervisors' ethical leadership inhibits academic misconduct of graduate students" (p.7), "We also found that supervisors' ethical leadership can prohibit students' academic misconduct through enhancing students' moral efficacy…."(p.23) are very misleading. When the authors said "supervisors' ethical leadership inhibits graduate students' academic misconduct", it should be "supervisors' ethical leadership reduces graduate students' agreement level (or acceptance level) of these misconducts," not their actual misconduct committed. Although the authors considered to use "projective techniques" to alleviate the concerns of respondents, I don't think the way items asked can justify or interpret attitudes to actual behaviors. In sum, the authors must clarify the measurement of variables and rewrite the whole article. 

Response: Thank you for your encouragement. We have tried our best to incorporate your suggestions and make our paper better.

We checked more literature and found that the measurement of academic misconduct in past research included attitudes (Jian, Marion, Wang, & Behavior, 2019), intention (Cronan, Mullins, & Douglas, 2018), and behaviors (Marsden, Carroll, & Neill, 2005). Since it is not easy to directly obtain data on academic misconduct, and respondents are generally unwilling to report unethical behaviors, we measured the attitude toward academic misconduct, that is, the acceptance of academic misconduct. In the introduction, we have clearly stated that our research object is attitudes toward academic misconduct. We have revised the wording throughout our paper.

R1-2 This paper has two sets of hypotheses. First set (study 1) has five hypotheses, and the second set (study 2) has four. I would suggest to renumber (such as H1-1, 1-2, 1-3, 2-1, 2-2…) these hypotheses for better clarity. If possible, the authors may provide graph(s) to visually show the set of five hypotheses and another set of four hypotheses.

Response: As you correctly pointed out, Study 1 tested all the hypotheses (H1-H5), while study 2 did not test the effect of the ethical climate (H5). Thank you for suggesting that we can distinguish the hypotheses tested by different studies in the naming: H1-1 indicates Hypothesis 1 in Study 1, and H1-2 indicates Hypothesis 1 in Study 2.

However, since the two studies tested the same hypotheses, giving different names to the hypotheses may lead readers to think that we are testing different hypotheses and increase the number of hypotheses (nine hypotheses seem too complex). We checked the articles in well-known academic journals, such as AMJ and JAP, that tested the same hypotheses using multiple methods. They named the hypotheses in a similar way to ours. Therefore, we retain the original naming. 

Following your suggestion, we have described the model and hypotheses in Figure 1 (see Pages 4-5 in the text).

R1-3 In the scale measuring “Ethical climate,” please explain more on why two items were deleted from the original 7-item scale. In my view, items “our team has policies regarding ethical behavior” and “our team enforces policies regarding ethical behavior” can be modified to fit the situation of the present study, such as “our research team has integrity policies regarding academic ethical behavior.”

Response: Thanks for your valuable comments. We used the scale of Schwepker Jr (2001). This measure was developed specifically to measure “the presence and enforcement of codes of ethics, corporate policies on ethics, and top management actions related to ethics” (p. 41). 

The original items in the scale are: (1) "our company has formal, written code of ethics," (2) "our company enforces a code of ethics," (3)"our company has policies regarding ethical behavior," (4)"our company enforces policies regarding ethical behavior," (5)"unethical behavior not tolerated," (6)"our company was reprimanded for behavior leading to personal gain," and (7) “our company was reprimanded for behavior leading to corporate gain."

Since the original scale was aimed at the corporate-level moral climate, we made appropriate adaptations to apply it in the academic context. However, the description of "policy" in the third and fourth items do not apply to our research background for two reasons: First, there are significant differences between developing and developed countries in formulating and implementing policies against academic misconduct. In most universities in China, although there are policy documents at the university level, these policies are often not strictly enforced and implemented (Gao, 2020; Li & Wang, 2019; Lu, 2019). In academic teams, the policies on academic misconduct are usually not strictly implemented. Second, we compared the scales in Chinese and English before the survey, as well as back-translated and compared them for semantic equivalence. Some graduate students filled out the translated scale and felt that these two items did not fit their daily study and life scenarios. We discussed this issue when designing the questionnaire and decided to delete these two items.

R1-4 This paper also needs a revision on its English writing. Some sentences are not complete, such as “Even the same ethics behaviors are not imitated equally by observers [10]. In theory, the social learning…” (p. 4). Some are typos or wrong use of words, such as “In high education, moral…” (p. 8), to name a few.

Response: Thank you very much for your valuable comments. We are sorry for these issues. We have proofread the paper and hope you find this updated version more readable.

Responses to Comments of Referee 2

Thank you for your detailed and constructive comments. Our responses to your comments (underlined in blue) are as follows.

R2-1 The paper reports a study trying to explain why and how supervisor gender and ethical leadership affects post-graduate students' social learning process. The authors adopted a multi-method approach to test the predictions in two studies. One study was a field experiment, and the second study was an experiment to enhance the robustness of the findings in the first study. The object of the analysis is presented in a convincing way and highlight its relevance. Although the paper is well written and the topic is certainly of interest, I am not sure it is addressed in the best way possible.

Response: Thank you for acknowledging the merits of our research. We have read your comments carefully and tried to improve the paper based on your comments as much as possible.

R2-1 In the first study (Study 1), they collected data from graduate students in four Chinese business schools. A total of 301 subjects correctly answered four questionaries, with a total of 26 items to measure Academic misconduct, Supervisor's ethical leadership, Ethical efficacy, and Ethical climate. Concerning the dataset, the authors should consider and comment on a possible percentage of fake answers of the questionaries. 

Response: You raised an excellent question. If the measured variables are sensitive, social desirability bias may occur. Social desirability bias may produce inaccurate measurements and can moderate, attenuate, or inflate the relationship between predictor and criterion variables (Dalton & Ortegren, 2011, p. 75). Typical methods to reduce social desirability bias include controlling for impression management (post-hoc approach) and indirect questioning (ex-ante approach) (Bossuyt & Van Kenhove, 2018).

We feel that social disability bias did not threaten the reliability of our conclusions for several reasons. First, we measured social disability in Study 1. We did not enter it into the equations because its Cronbach alpha coefficient did not exceed 0.8. Since it is acceptable that Cronbach's alpha exceeds 0.7 (Streiner, 2003), we now include social disability as a control variable. The updated regression results still support the hypotheses. Second, we adopted an indirect questioning method to measure attitudes toward academic misconduct in Study 1. Third, the survey in Study 2 was conducted online. Anonymous online answers can reduce social disability bias. To sum up, these measures can effectively prevent social disability bias.

Following your suggestions, we have added a discussion of social disability bias in the Discussion section (see Page 24 in the text).

“The measures of attitudes are susceptible to social desirability bias [69]. Although our research adopted the indirect questioning method and added a control variable of social desirability, those measures could not wholly prevent respondents from providing socially desirable responses. Future researchers can use big data and management information systems to directly measure academic misconduct (e.g., cheating, plagiarism). In that case, the measurement of academic misconduct will be more direct and accurate.”

R2-3 In addition, it might be interesting to know if the supervisors are also the directors of their academic team. Moreover, the authors should describe the code for the age variable as they described the other control variables in the paper.

Response: Thank you for reminding us of this important issue. In different countries, the design of the tutorial system is different. For example, in addition to an academic supervisor, students at Cambridge University in the United Kingdom also have a supervisor who is responsible for solving non-academic related problems. 

According to your comments, we have supplemented relevant information in the sample introduction in Study 1. The scope of the academic team we investigated was limited to doctoral students, academic masters, and professional masters. The primary responsibility of the supervisor was to provide academic guidance. Their qualification as a supervisor was assessed annually according to their academic achievements. The average number of students in the academic team was 7.6, with a minimum size of 3 and a maximum size of 15 (see Page 11 in the text).

We have added the coding information for students' age: 1 for students under 22, 2 for those 22-25, 3 for those 26-29, and 4 for those over 30 (see Page 13 in the text).

R2-4 Concerning the methodology used in this study, the authors used parametric tests such as Pearson's test, a method of statistical analysis that assume a normal distribution. In my opinion, it would be better to use nonparametric tests to avoid this assumption. 

Response: Thank you. We agree with you that most of the statistical methods we adopted require that the data follow a normal distribution. 

We made Q-Q plots for the data distributions of the dependent variables in Study 1 and Study 2. As can be seen from the Q-Q plots , the dependent variables in the two studies did not deviate significantly from the normal distribution.

Because many empirical data could not meet the typical normal distribution, scholars discussed this issue. For example, Li et al. (2012, p. 3082) proposed: “We suggest there is a common misconception of the need to meet the ‘normality assumption’ in linear regression techniques, and the validity of performing linear regression is compromised when this assumption is violated…In a large sample, the use of a linear regression technique, even if the dependent variable violates the ‘normality assumption’ rule, remains valid”(p.3082).

In Study 1, we used HLM analysis. The assumptions of this statistical method are (1) linearity, (2) homogeneity of variance, and (3) the residuals of the model are normally distributed (Garson, 2013). Since the first study used a large sample, the assumption of the normal distribution can be relaxed. The regression results we reported were “the final estimation of fixed effects with robust standard errors," which relaxed the assumption of normal distribution (Garson, 2013). In addition, we performed the test of homogeneity of variance in HLM8.0 software, and the results showed that the variances were homogeneous (�2(59) = 25.58, p > .50). Therefore, the statistical methods in Study 1 are appropriate.

Study 2 had small samples in each scenario. Thus the normal distribution was not easily met. We now use the Mann-Whitney U Test according to your suggestion (see the results on Page 19 in the text). In study 2, we also used ANOVA and Hayes' process macro to test the mediation and moderation effects. The ANOVA is relatively robust against violations of the normality assumption (Schmider et al., 2010). The bootstrap in Hayes' process macro does care if the data is normally distributed (Hayes, Montoya, & Rockwood, 2017). 

R2-5 In table 2, the authors should include the standard errors and a third or fourth decimal in all coefficients instead of writing 0.00 coefficient in models 4 and 6. Furthermore, in the title of Table 2, the regression of the ethical climate model does not appear.

Response: Thank you for your suggestions. We have updated Table 2, and the results include four decimal places. We have modified the title of Table 2.

R2-6 As they said in the paper, given that gender was unbalanced in Study 1, they conducted a lab experiment (Study 2) to enrich their results and provide evidence of causality. However, in my opinion (if I understood correctly), the experiment was not designed as well as possible or a thorough experimental design section is needed in the paper. 

Response: Thank you for your suggestion. We have reviewed more literature, and we now describe our second study as a “vignette study” (Atzmüller & Steiner, 2010; Nivette, Nägel, & Stan, 2022), that is, experiments in survey research (Steiner, Atzmüller, & Su, 2017).

Vignette studies combine the advantages of experiment and survey: "Vignette studies combine ideas from classical experiments and survey methodology to counterbalance each approach's weakness. It allows researchers to construct a scenario describing a particular situation while systematically varying key characteristics that are hypothesized to influence the outcome" (Atzmüller & Steiner, 2010, p. 128). In the past, many articles published in well-known academic journals (e.g., JAP, AMJ, OBHDP, JBE) have adopted similar research methods (e.g., Brian & Lindsey, 2013; Chen, Chen, & Sheldon, 2016; Ji, Huang, & Li, 2021; Mell, DeChurch, Leenders, & Contractor, 2020; Zhu, 2015).

R2-6 As a control experiment, the authors should be considered the same control variables of the subjects as in the first study. Moreover, the authors should control the number of subjects in the four groups considering high (low) ethical leadership and male (female) supervisor.

Response: Study 2 adopted random assignment of treatment. Since alternative causes were randomly distributed across conditions, they became perfectly balanced. Random assignment does not need control variables (Cook & Campbell, 1979; Harris, 2008) unless the experiment contains confounding variables. Confounding variables are variables that coincide with different levels of the independent variable. In Study 2, we could not think of any confounding variables; thus, we did not add control variables. The sample sizes in each condition did not vary significantly (from 35 to 48 participants). Theoretically, the sample size would not be a confounding variable that coincides with our manipulations.

We reviewed studies published in AMJ and JAP. Studies using vignette experiments did not include control variables in the research design (e.g., Mayer, Ong, Sonenshein, & Ashford, 2019; McCarthy & Levin, 2019; Park, Tangirala, Hussain, & Ekkirala, 2022; Speach, Badura, & Blum, 2022; Xu, Loi, & Chow, 2022).

R2-5 In addition, in order to control the fake answers of the questionaries, an experimental design with real reward could be considered. In fact, it would be interesting to include some game that implements a punishment system as a solution to the ethical issues.

Response: Thank you for your comments. Because our vignette study was online and anonymous, social disability bias could be reduced. Your suggestions for using games in the experiment are valuable. Since our research focuses on attitudes rather than behaviors, we used self-report to measure the dependent variable. Inspired by your comments, we discuss the methodological limitations of the second study:

“Furthermore, one of the main shortcomings of the vignette study is the artificial nature of the research situation. Although we adopted a combination of texts and pictures to immerse the subjects in the scenarios as much as possible, those approaches might be insufficient to generate strong incentives in experiments. To engage participants' senses more deeply, future researchers can use more advanced technologies, such as videos, games, and virtual reality technology, to let participants see and feel the appearance and behaviors of the supervisors” (see Page 24 in the text).

R2-6 Concerning the methodology used in the study 2, again the authors used parametric tests. I would use a Mann-Whitney test instead of t-test.

Response: Thank you for your suggestion. We have used the Mann-Whitney test in Study 2 based on your suggestion. Please also see our response to your Comment R2-4.

R2-7 Minor comments:

- In page 11, in Hypothesis 3, instead of writing “Supervisor gender moderates…” I could write “Woman (or female) supervisor…”

Response: We have made modifications.

“H3: The relationship between ethical leadership and graduate students' moral efficacy is stronger for female supervisors than for male supervisors” (see Page 9 in the text).

- In my opinion, the authors should write the name of the indices that they used in the analysis of the variables. For example, The internal consistency (�), the Intraclass correlation coefficient (ICC), the measure of within-group agreement, …

Response: Thank you for your comments. According to your suggestion, we have given the full name of the abbreviated professional terms when they first appear. If the same professional terms appear later, we use abbreviations.

- The authors should check a number of cites in the text. In page 13, "…Zhang and Yu [55]…" is [56]. In page 14, "(Bliese, 2000)" in reference [57]. "The scale developed by Schwepker Jr [56]…" is [58]. Page 27, "…Calhoun, 1995…" in reference [67]

Response: Thanks for pointing out errors in our typography. We have proofread the references. We have made revisions according to the journal's bibliography requirements.

- The authors should check reference Zhang and Yu 2017.

Response: Thank you for your detailed advice. We have revised the references.

- In page 16, “An EFA was performed…” include Explanatory Factor Analysis (EFA)

Response: Thank you for your comments. We have modified it.

- References 65-66 are not cited in the text

Response: We are sorry for the negligence when using Endnote for typesetting. We have made corrections.

R2-8 My overall impression of the manuscript is that it needs a thorough revision following the comments made in the preceding lines. In particular, the lab experimental design. I rather suggest to send it a specific journal.

Response: Thank you for your comments. We have revised the paper as much as possible. We hope that the revised version can meet the academic requirement of the journal and get your approval.

The research scope of PLOS ONE is diverse. We checked the articles published by PLOS ONE in the last five years. In terms of research topics, there are articles about ethical leadership (e.g., Chaman, Zulfiqar, Shaheen, & Saleem, 2021; Gigol, 2021; Grego-Planer, 2022; Zhang & Yao, 2019) and academic misconduct (Abdulghani et al., 2019; Baran & Jonason, 2020; Stavale et al., 2019; Suart, Neuman, & Truant, 2022). In terms of research methods, some studies used vignette studies similar to the research design of ours (e.g., Abdul Aziz, Flanders, & Nungsari, 2022; Aharoni, Kleider-Offutt, Brosnan, & Fernandes, 2020; Blum et al., 2019; Koens, Strauß, Klein, Schafer, & von dem Knesebeck, 2022; Reed, Meyer, Okun, Best, & Hooley, 2020; Rousseau, Rozenberg, Perrodeau, & Ravaud, 2018).

We will try our best to modify our paper to meet the publication requirements of the journal.

References

[1] Abdul Aziz, N. I., Flanders, S., & Nungsari, M. 2022. Social expectations and government incentives in Malaysia's COVID-19 vaccine uptake. PloS one, 17(9): e0275010.

[2] Abdulghani, H. M., Haque, S., Almusalam, Y. A., Alanezi, S. L., Alsulaiman, Y. A., Irshad, M., & Khamis, N. 2019. Self-reported cheating among medical students: An alarming finding in a cross-sectional study from Saudi Arabia (vol 13, e0194963, 2018). Plos One, 14(4): e0215862.

[3] Aharoni, E., Kleider-Offutt, H. M., Brosnan, S. F., & Fernandes, S. 2020. Slippery scales: Cost prompts, but not benefit prompts, modulate sentencing recommendations in laypeople. Plos One, 15(7): e023676.

[4] Atzmüller, C. & Steiner, P. M. 2010. Experimental vignette studies in survey research. Methodology: European Journal of Research Methods for the Behavioral Social Sciences, 6(3): 128-138.

[5] Baran, L. & Jonason, P. K. 2020. Academic dishonesty among university students: The roles of the psychopathy, motivation, and self-efficacy. Plos One, 15(8): e0238141.

[6] Blum, R. W., Sheehy, G., Li, M., Basu, S., El Gibaly, O., Kayembe, P., Zuo, X., Ortiz, J., Chan, K. S., & Moreau, C. 2019. Measuring young adolescent perceptions of relationships: A vignette-based approach to exploring gender equality. Plos One, 14(6): e0218863.

[7] Bossuyt, S. & Van Kenhove, P. 2018. Assertiveness Bias in Gender Ethics Research: Why Women Deserve the Benefit of the Doubt. Journal of Business Ethics, 150(3): 1-13.

[8] Brian, G. W. & Lindsey, N. G. 2013. The Antecedents of Moral Imagination in the Workplace: A Social Cognitive Theory Perspective. Journal of Business Ethics, 114(1): 61-73.

[9] Chaman, S., Zulfiqar, S., Shaheen, S., & Saleem, S. 2021. Leadership styles and employee knowledge sharing: Exploring the mediating role of introjected motivation. Plos One, 16(9): e0257174.

[10] Chen, M., Chen, C. C., & Sheldon, O. J. 2016. Relaxing moral reasoning to win: How organizational identification relates to unethical pro-organizational behavior. Journal of Applied Psychology, 101(8): 1082-1096.

[11] Cook, T. D. & Campbell, D. T. 1979. The design and conduct of true experiments and quasi-experiments in field settings. In R. T. Mowday & R. M. Steers (Eds.), Reproduced in part in Research in Organizations: Issues and Controversies: Goodyear Publishing Company.

[12] Cronan, T. P., Mullins, J. K., & Douglas, D. E. 2018. Further understanding factors that explain freshman business students’ academic integrity intention and behavior: Plagiarism and sharing homework. Journal of Business Ethics, 147(1): 197-220.

[13] Dalton, D. & Ortegren, M. J. J. o. B. E. 2011. Gender differences in ethics research: The importance of controlling for the social desirability response bias. 103(1): 73-93.

[14] Gao, Y. 2020. [Syetem improvement for governing academic misconduct of postgraduates: Based on the investigation of academic norms text for C9 universities]. Modern Education Science, 4: 38-43.

[15] Garson, G. D. 2013. Hierarchical linear modeling: Guide and applications. London, UK: Sage.

[16] Gigol, T. 2021. Leadership, religiousness, state ownership of an enterprise and unethical pro-organizational behavior: The mediating role of organizational identification. Plos One, 16(5): e0251465.

[17] Grego-Planer, D. 2022. The relationship between benevolent leadership and affective commitment from an employee perspective. Plos One, 17(3): e0264142.

[18] Harris, P. 2008. Designing and reporting experiments in psychology: McGraw-Hill Education (UK).

[19] Hayes, A. F., Montoya, A. K., & Rockwood, N. J. 2017. The analysis of mechanisms and their contingencies: PROCESS versus structural equation modeling. Australasian Marketing Journal, 25(1): 76-81.

[20] Ji, J., Huang, Z., & Li, Q. 2021. Guilt and corporate philanthropy: the case of the privatization in china. Academy of Management Journal, 64(6): 1969-1995.

[21] Jian, H., Marion, R., Wang, W. J. E., & Behavior. 2019. Academic integrity from China to the United States: The acculturation process for Chinese graduate students in the United States. Ethics & Behavior, 29(1): 51-70.

[22] Koens, S., Strauß, A., Klein, J., Schafer, I., & von dem Knesebeck, O. 2022. Public perceptions of urgency of severe cases of COVID-19 and inflammatory gastrointestinal disease. PloS one, 17(8): e0273000.

[23] Li, X., Wong, W., Lamoureux, E. L., & Wong, T. Y. 2012. Are linear regression techniques appropriate for analysis when the dependent (outcome) variable is not normally distributed? Investigative ophthalmology visual science, 53(6): 3082-3083.

[24] Li, X. & Wang, L. L. 2019. [System construction and improvement for governing academic misconduct of postgraduates: Based on the document text analysis of the universities of Project 985]. Journal of Graduate Education, 3: 53-59.

[25] Lu, X. F. 2019. [A comparative study on the institutions of preventing and dealing with academic misconduct in domestic and foreign universities]. China University Science & Technology, 9: 12-16.

[26] Marsden, H., Carroll, M., & Neill, J. T. 2005. Who cheats at university? A self‐report study of dishonest academic behaviours in a sample of Australian university students. Australian Journal of Psychology, 57(1): 1-10.

[27] Mayer, D. M., Ong, M., Sonenshein, S., & Ashford, S. 2019. The money or the morals? When moral language is more effective for selling social issues. Journal of Applied Psychology, 104(8): 1058-1076.

[28] McCarthy, J. E. & Levin, D. Z. 2019. Network residues: The enduring impact of intra-organizational dormant ties. Journal of Applied Psychology, 104(11): 1434-1445.

[29] Mell, J. N., DeChurch, L. A., Leenders, R. T. A. J., & Contractor, N. 2020. Identity asymmetries: An experimental investigation of social identity and information exchange in multiteam systems. Academy of Management Journal, 63(5): 1561-1590.

[30] Nivette, A., Nägel, C., & Stan, A. 2022. The use of experimental vignettes in studying police procedural justice: A systematic review. Journal of Experimental Criminology, Forthcoming.

[31] Park, H., Tangirala, S., Hussain, I., & Ekkirala, S. 2022. How and when managers reward employees’ voice: The role of proactivity attributions. Journal of Applied Psychology: Forthcoming.

[32] Reed, L. I., Meyer, A. K., Okun, S. J., Best, C. K., & Hooley, J. M. 2020. In smiles we trust? Smiling in the context of antisocial and borderline personality pathology. Plos One, 15(6): e0234574.

[33] Rousseau, A., Rozenberg, P., Perrodeau, E., & Ravaud, P. 2018. Variation in severe postpartum hemorrhage management: A national vignette-based study. Plos One, 13(12): e0209074.

[34] Schmider, E., Ziegler, M., Danay, E., Beyer, L., Bühner, M. J. M. E. J. o. R. M. f. t. B., & Sciences, S. 2010. Is it really robust? Reinvestigating the robustness of ANOVA against violations of the normal distribution assumption. 6(4): 147-151.

[35] Schwepker Jr, C. H. 2001. Ethical climate's relationship to job satisfaction, organizational commitment, and turnover intention in the salesforce. Journal of Business Research, 54(1): 39-52.

[36] Speach, M. E. P., Badura, K. L., & Blum, T. C. 2022. Everything is negotiable, but not for everyone: The role of disability in compensation. Journal of Applied Psychology: Forthcoming.

[37] Stavale, R., Ferreira, G. I., Martins Galvao, J. A., Zicker, F., Carvalho Garbi Novaes, M. R., de Oliveira, C. M., & Guilhem, D. 2019. Research misconduct in health and life sciences research: A systematic review of retracted literature from Brazilian institutions. Plos One, 14(4): e0214272.

[38] Steiner, P. M., Atzmüller, C., & Su, D. 2017. Designing valid and reliable vignette experiments for survey research: A case study on the fair gender income gap. Journal of Methods Measurement in the Social Sciences, 7(2): 52-94.

[39] Streiner, D. L. 2003. Starting at the beginning: An introduction to coefficient alpha and internal consistency. Journal of Personality Assessment, 80(1): 99-103.

[40] Suart, C., Neuman, K., & Truant, R. 2022. The impact of the COVID-19 pandemic on perceived publication pressure among academic researchers in Canada. PloS one, 17(6): e0269743.

[41] Xu, A. J., Loi, R., & Chow, C. W. 2022. Does taking charge help or harm employees’ promotability and visibility? An investigation from supervisors’ status perspective. Journal of Applied Psychology, Forthcoming.

[42] Zhang, X. & Yao, Z. 2019. Impact of relational leadership on employees' unethical pro-organizational behavior: A survey based on tourism companies in four countries. Plos One, 14(12): e0225706.

[43] Zhu, Y. X. 2015. The role of Qing (positive emotions) and Li (rationality) in Chinese entrepreneurial decision making: A Confucian Ren-Yi wisdom perspective. Journal of Business Ethics, 126(4): 613-630.

---

## [Decision Letter · Decision Letter 1]

20 Jan 2023

PONE-D-22-10477R1Supervisors' ethical leadership and graduate students' attitudes toward academic misconductPLOS ONE

Dear Dr. Zhang,

Thank you for submitting your manuscript to PLOS ONE. After careful consideration, we feel that it has merit but does not fully meet PLOS ONE’s publication criteria as it currently stands. Therefore, we invite you to submit a revised version of the manuscript that addresses the points raised during the review process. Both reviewers and myself see great improvement in your manuscript. Please make sure to incorporate the suggestions of the reviewer in the revised version and I will take the final decision without sending out to reviewers again.

We look forward to receiving your revised manuscript.

Kind regards,

Iván Barreda-Tarrazona, PhD

Academic Editor

PLOS ONE

Journal Requirements:

Reviewers' comments:

Reviewer's Responses to Questions

**Comments to the Author**

1. If the authors have adequately addressed your comments raised in a previous round of review and you feel that this manuscript is now acceptable for publication, you may indicate that here to bypass the “Comments to the Author” section, enter your conflict of interest statement in the “Confidential to Editor” section, and submit your "Accept" recommendation.

Reviewer #1: All comments have been addressed

Reviewer #2: All comments have been addressed

2. Is the manuscript technically sound, and do the data support the conclusions?

Reviewer #1: Yes

Reviewer #2: Yes

3. Has the statistical analysis been performed appropriately and rigorously? 

Reviewer #1: Yes

Reviewer #2: Yes

4. Have the authors made all data underlying the findings in their manuscript fully available?

Reviewer #1: Yes

Reviewer #2: Yes

5. Is the manuscript presented in an intelligible fashion and written in standard English?

Reviewer #1: Yes

Reviewer #2: Yes

6. Review Comments to the Author

Reviewer #1: The authors have made substantial revisions and solved the fundamental question whether attitudes or behaviors were measured and analyzed. Moreover, the writing was improved to be more readable.

Reviewer #2: In the present version, the authors implemented most of the reviewers’ suggestions.

There are some points in the new text which should be checked by the authors for language and editing.

Table 2, note (3). In my opinion, it is better to write “*** Coefficient significant at 1%, ** significant at 5%, * significant at 10%”. And the authors should add this note in table 1.

7. PLOS authors have the option to publish the peer review history of their article (what does this mean?). If published, this will include your full peer review and any attached files.

Reviewer #1: No

Reviewer #2: No

---

## [Author Response · Author response to Decision Letter 1]

17 Feb 2023

Dear reviewers,

In the last draft, we responded to all your concerns and comments. We solved the weaknesses in theory and methods to make significant improvements. Thank you very much for your recognition of our modification.

Therefore, in this updated draft, we mainly corrected the minor problems. We have modified the notes of Table 1 and Table 2 according to your comments. We have adjusted the format of Figure 1 according to the journal’s requirements. We have checked the writing and grammar. 

Thank you again for taking the precious time to help us improve this study. 

Best, 

The authors' team

---

## [Editor Report · Decision Letter 2]

1 Mar 2023

Supervisors' ethical leadership and graduate students' attitudes toward academic misconduct

PONE-D-22-10477R2

Dear Dr. ZHANG,

We’re pleased to inform you that your manuscript has been judged scientifically suitable for publication and will be formally accepted for publication once it meets all outstanding technical requirements.

Kind regards,

Iván Barreda-Tarrazona, PhD

Academic Editor

PLOS ONE
---

## [Editor Report · Acceptance letter]

31 Mar 2023

PONE-D-22-10477R2 

Supervisors' ethical leadership and graduate students' attitudes toward academic misconduct 

Dear Dr. ZHANG:

I'm pleased to inform you that your manuscript has been deemed suitable for publication in PLOS ONE. Congratulations! Your manuscript is now with our production department. 

Kind regards, 

on behalf of

Dr. Iván Barreda-Tarrazona 

Academic Editor

PLOS ONE